# Fair Federated Learning via Bounded Group Loss

## Abstract

Fair prediction across protected groups is an important constraint for many federated learning applications. However, prior work studying group fair federated learning lacks formal convergence or fairness guarantees. In this work we propose a general framework for provably fair federated learning. In particular, we explore and extend the notion of Bounded Group Loss as a theoretically-grounded approach for group fairness. Using this setup, we propose a scalable federated optimization method that optimizes the empirical risk under a number of group fairness constraints. We provide convergence guarantees for the method as well as fairness guarantees for the resulting solution. Empirically, we evaluate our method across common benchmarks from fair ML and federated learning, showing that it can provide both fairer and more accurate predictions than baseline approaches.

## 1 Introduction

Group fairness aims to mitigate unfair biases against certain protected demographic groups (e.g. race, gender, age) in the use of machine learning. Many methods have been proposed to incorporate group fairness constraints in centralized settings (e.g., Agarwal et al., 2018; Feldman et al., 2015; Hardt et al., 2016; Zafar et al., 2017a). However, there is a lack of work studying these approaches in the context of federated learning (FL), a training paradigm where a model is fit to data generated by a set of disparate data silos, such as a network of remote devices or collection of organizations (Kairouz et al., 2019; Li et al., 2020; McMahan et al., 2017). Mirroring concerns around fairness in non-federated settings, many FL applications similarly require performing fair prediction across protected groups. Unfortunately, as we show in Figure 1, naively applying existing approaches to each client in a federated network in isolation may be inaccurate due to heterogeneity across clients—failing to produce a fair model across the entire population (Zeng et al., 2021).

Several recent works have considered addressing this issue by exploring specific forms of group fairness in FL (e.g., Chu et al., 2021; Cui et al., 2021; Du et al., 2021; Papadaki et al., 2022; Rodríguez-Gálvez et al., 2021; Zeng et al., 2021). Despite promising empirical performance, these prior works lack formal guarantees surrounding the resulting fairness of the solutions (Section 2), which is problematic as it is unclear how the methods may perform in real-world FL deployments.

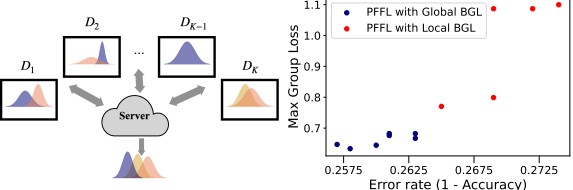

**Figure 1:** Naively applying fair learning method locally at each client might be problematic. Left: Due to data heterogeneity in FL, data distributions conditioned on each protected attribute (shown in different colors) may differ across clients. Fair FL aims to learn a model that provides fair prediction on the entire data distribution. Right: Empirical results (ACS dataset) verify that training with local fairness constraints alone induces higher error and worse fairness than using a global fairness constraint.

In this work we provide a formulation and method for group fair FL that can provably satisfy global fairness constraints. Common group fairness notions that aim to achieve equal prediction quality between any two protected groups (e.g., Demographic Parity, Equal Opportunity (Hardt et al., 2016)) are difficult to provably satisfy while simultaneously finding a model with high utility. Instead, we consider a different fairness notion known as *Bounded Group Loss (BGL)* (Agarwal et al., 2019), which aims to promote worst group's performance, to capture these common group fairness criteria. As we show, a benefit of this approach is that in addition to

having practical advantages in terms of fairness-utility trade-offs (Section 5), it maintains smoothness and convexity properties that can equip our solver with favorable theoretical guarantees.

Based on our group fairness formulation, we then provide a scalable method (PFFL) to solve the proposed objectives via federated saddle point optimization. Theoretically, we provide convergence guarantees for the method as well as fairness and generalization guarantees for the resulting solutions. Empirically, we demonstrate the effectiveness of our approach on common benchmarks from fair machine learning and federated learning. We summarize our main contributions below:

- We propose a novel fair federated learning framework for a range of group fairness notions. Our framework models the fair FL problem as a saddle point optimization problem and leverages variations of Bounded Group Loss (Agarwal et al., 2019) to capture common forms of group fairness. We also extend BGL to consider a new fairness notion called Conditional Bounded Group Loss (CBGL), which may be of independent interest and utility in non-federated settings.
- We propose a scalable federated optimization method for our group fair FL framework. We provide a regret bound analysis for our method under convex ML objectives to demonstrate formal convergence guarantees. Further, we provide fairness and generalization guarantees on the model for a variety of fairness notions.
- Finally, we evaluate our method on common benchmarks used in fair machine learning and federated learning. In all settings, we find that our method can significantly improve model fairness compared to baselines without sacrificing model accuracy. Additionally, even though we do not directly optimize classical group fairness constraints (e.g., Demographic Parity, Equal Opportunity), we find that our method can still provide comparable/better fairness-utility trade-offs relative to existing approaches when evaluated on these metrics.

## 2 BACKGROUND AND RELATED WORK

**Fair Machine Learning.** Algorithmic fairness in machine learning aims to identify and correct bias in the learning process. Common approaches for obtaining fairness include pre-processing methods that rectify the features or raw data to enhance fairness (Calmon et al., 2017; Feldman et al., 2015; Zemel et al., 2013); post-processing methods that revise the prediction score for a trained model (Dwork et al., 2018; Hardt et al., 2016; Menon & Williamson, 2018); and in-processing methods that directly modify the training objective/solver to produce a fair predictor (Agarwal et al., 2018; 2019; Woodworth et al., 2017; Zafar et al., 2017a;b). Most existing methods in fair ML rely on using a centralized dataset to train and evaluate the model. As shown in Figure 1, in the federated setting where data is privately distributed across different data silos, directly applying these methods locally only ensures fairness for each silo rather than the entire population. Developing effective and efficient techniques for fair FL is thus an important area of study.

**Fair Federated Learning.** In FL, definitions of fairness may take many forms. A commonly studied notion of fairness is representation parity (Hashimoto et al., 2018), whose application in FL requires the model's performance across all clients to have small variance (Donahue & Kleinberg, 2021; Li et al., 2019a; 2021; Mohri et al., 2019; Yue et al., 2021). In this work we instead focus on notions of *group fairness*, in which every data point in the federated network belongs to some (possibly) protected group, and we aim to find a model that doesn't introduce bias towards any group.

Recent works have proposed various objectives for group fairness in federated learning. Zeng et al. (2021) proposes a bi-level optimization objective that minimizes the difference between each group's loss while finding an optimal global model. Similarly, several works propose using a constrained optimization problem that aims to find the best model subject to an upper bound on the group loss difference (Chu et al., 2021; Cui et al., 2021; Du et al., 2021; Rodríguez-Gálvez et al., 2021). Different from these approaches, our method focuses on a fairness constraint based on upperbounding the loss of each group with a constant rather than the loss difference between any two groups. More closely related to our work, Papadaki et al. (2022) weighs the empirical loss given each group by a trainable vector $\lambda$ and finds the best model for the worst case $\lambda$. Though similar to our method for $\zeta = 0$, this approach fails to achieve both strong utility and fairness performance under non-convex loss functions (see Section 5). Zhang et al. (2021) also propose a similar objective to learn a model with unified fairness. Among these works, Zeng et al. (2021) and Cui et al. (2021) also provide simplified convergence and fairness guarantees for their method. However, these works lack formal analyses around the convergence for arbitrary convex loss functions as well as the behavior of the fairness constraint over the true data distribution. Ours is the first work we are aware to provide such guarantees in the context of group fair federated learning.

## 3 FAIR FEDERATED LEARNING VIA BOUNDED GROUP LOSS

In this section we first formalize the group fair federated learning problem and a fairness-aware objective solving this problem (Section 3.1). We then provide several examples of group fairness based on the notion of BGL and show how to incorporate them into our framework (Section 3.2).

### 3.1 SETUP: GROUP FAIR FEDERATED LEARNING

Many applications of FL require treatment of data belonging to protected groups (e.g., race, gender, age). This is particularly common in applications of cross-silo FL, where we may wish to produce a model that fairly treats individuals from various demographic groups across a collection of data silos (e.g. hospitals, schools, financial institutions) (Chu et al., 2021; Cui et al., 2021; Vaid et al., 2021).

**FL Setup.** Following standard federated learning scenarios (McMahan et al., 2017), we consider a network with $K$ different clients. Each client $k \in [K]$ has access to training data $\hat{\mathcal{D}}_k := \{(x_i, y_i, a_i)\}_{i=1,\cdots,m_k}$ sampled from the true data distribution $\mathcal{D}_k$, where $x_i$ is an observation, $y_i \in Y$ is the label, $a_i \in A$ is the protected attribute. Let the hypothesis class be $\mathcal{H}$ and for any model $h \in \mathcal{H}$, and define the loss function on data $(x, y, a)$ to be $l(h(x), y)$. Federated learning applications typically aim to solve:

$$\min_{h \in \mathcal{H}} \mathcal{F}(h) = \min_{h \in \mathcal{H}} \mathbb{E}_{(x,y) \sim \mathcal{D}} \left[ l(h(x), y) \right]. \tag{1}$$

In practice, $\mathcal{D}_k$ is estimated by observing $\{(x_i, y_i, a_i)\}_{i=1,\cdots,m_k}$, and we solve the empirical risk:

$$\min_{h \in \mathcal{H}} F(h) = \min_{h \in \mathcal{H}} \frac{1}{K} \sum_{k=1}^{K} \frac{1}{m_k} \sum_{i=1}^{m_k} l(h(x_{k,i}), y_{k,i}). \tag{2}$$

For simplicity, we define $f_k(h) = \frac{1}{m_k} \sum_{i=1}^{m_k} l(h(x_{k,i}), y_{k,i})$ as the local objective for client $k$. Further, we assume $h$ is parameterized by a vector $w \in \mathbb{R}^p$ where $p$ is the number of parameters. We will use $F(w)$ and $f_k(w)$ to represent $F(h)$ and $f_k(h)$ intermittently in the remainder of the paper.

**Fairness via Constrained Optimization.** When a centralized dataset is available, a standard approach to learn a model that incorporates fairness criteria is to solve a constrained optimization problem where one tries to find the best model subject to some fairness notion (Agarwal et al., 2019; Barocas et al., 2019). Following this formulation, we formalize a similar learning problem in the federated setting, solving:

$$\min_{h \in \mathcal{H}} F(h) \quad \text{subject to } \mathbf{R}(h) \leq \boldsymbol{\zeta}, \tag{3}$$

where $\mathbf{R}(h), \boldsymbol{\zeta} \in \mathbb{R}^Z$ encodes the constraint set on $h$. For instance, the $z$-th constraint could be written as $\mathbf{R}_z(h) \leq \zeta_z$ where $\zeta_z$ is a fixed constant. This formulation is commonly used to satisfy group fairness notions such as equal opportunity, equalized odds (Hardt et al., 2016), and minimax group fairness (Diana et al., 2021).

To solve the constrained optimization problem 3, a common method is to use Lagrangian multipliers. In particular, let $\boldsymbol{\lambda} \in \mathbb{R}_+^Z$ be a dual variable and assume $\boldsymbol{\lambda}$ has $\| \cdot \|_1$ at most $B$. The magnitude of $B$ could be viewed as the regularization strength for the constraint term. Objective equation 3 can then be converted into the following saddle point optimization problem:

$$\min_{w} \max_{\boldsymbol{\lambda} \in \mathbb{R}_+^Z, \|\boldsymbol{\lambda}\|_1 \leq B} G(w; \boldsymbol{\lambda}) = \beta F(w) + \boldsymbol{\lambda}^T \mathbf{r}(w), \qquad \text{(Main Objective)}$$

where the $q$-th index of $\mathbf{r}$ encodes the $q$-th constraint from $\mathbf{R}$ (i.e. $\mathbf{r}_q(w) := \mathbf{R}_q(w) - \zeta_q$) and $\beta$ is a fixed constant. In other words, the objective finds the best model under the scenario where the fairness constraint is most violated (i.e., the regularization term is maximized).

There are two steps needed in order to provide meaningful utility and fairness guarantees for the model found by solving Main Objective: (1) showing that it is possible to recover a solution close to the *'optimal'* solution, (2) providing an upper bound for both the risk ($F(w)$) and the fairness constraint ($\mathbf{r}(w)$) given this solution. To formally define what an *'optimal'* solution is, in this work we aim to identify constraints that satisfy the following key assumption:

**Assumption 0 (Convexity of $G$).** *Assume that $G(w; \boldsymbol{\lambda})$ is convex in $w$ for any fixed $\boldsymbol{\lambda}$.*

**Remark.** In particular, since $G$ is linear in $\boldsymbol{\lambda}$, given a fixed $w_0$, we can find a solution to the problem $\max_{\boldsymbol{\lambda}} G(w_0; \boldsymbol{\lambda})$, denoted as $\boldsymbol{\lambda}^*$, i.e. $G(w_0; \boldsymbol{\lambda}^*) \geq G(w_0; \boldsymbol{\lambda})$ for all $\boldsymbol{\lambda}$. When $G$ is convex in $w$, we can argue that given a fixed $\boldsymbol{\lambda}_0$, there exists $w^*$ that satisfies $w^* = \arg\min_w G(w; \boldsymbol{\lambda}_0)$, i.e. $G(w^*; \boldsymbol{\lambda}_0) \leq G(w; \boldsymbol{\lambda}_0)$ for all $w$. Therefore, $(w^*, \boldsymbol{\lambda}^*)$ is a saddle point of $G(\cdot; \cdot)$, which is denoted as the optimal solution in our setting.

### 3.2 Formulating Fair FL: Bounded Group Loss and Variants

Many prior works in fair federated learning consider instantiating $\mathbf{R}(h)$ in equation 3 as a constraint that bounds the difference between any two groups' losses, a common technique used to enforce group fairness notions such as equalized odds and demographic parity (e.g., Chu et al., 2021; Cui et al., 2021; Zeng et al., 2021). Unfortunately, this results in $G(w; \boldsymbol{\lambda})$ becoming nonconvex in $w$, thus violating our Assumption 0. This nonconvexity is problematic as it increases the likelihood that a solver will find a local minima that either does not satisfy the fairness constraint or achieves poor utility. Instead of enforcing equity between the prediction quality of any two groups, in this work we explore using a constraint based on *Bounded Group Loss (BGL)* (Agarwal et al., 2019) which promotes worst group's prediction quality and propose new variants that can retain convexity assumptions while satisfying meaningful fairness notions. In particular, we explore three instantiations of group fairness constraints $\mathbf{R}(h)$ that satisfy Assumption 0 below.

**Instantiation 1 (Bounded Group Loss).** We begin by considering fairness via the Bounded Group Loss (defined below), which was originally proposed by Agarwal et al. (2019). Different from applying Bounded Group Loss in a centralized setting, BGL in the context of federated learning requires that for any group $a \in A$, the average loss for *all* data belonging to group $a$ is below a certain threshold. As we discuss in Section 4 this (along with general constraints of FL such as communication) necessitates the development of novel solvers and analyses for the objective.

**Definition 1** (Agarwal et al. (2019)). *A classifier $h$ satisfies Bounded Group Loss (BGL) at level $\zeta$ under distribution $\mathcal{D}$ if for all $a \in A$, we have $\mathbb{E}\left[l(h(x), y) | A = a\right] \leq \zeta$.*

In practice, we could define empirical bounded group loss constraint at level $\zeta$ under the empirical distribution $\widehat{\mathcal{D}} = \frac{1}{K} \sum_{k=1}^{K} \widehat{\mathcal{D}}_k$ to be

$$\mathbf{r}_a(h) := \sum_{k=1}^{K} \mathbf{r}_{a,k}(h) = \sum_{k=1}^{K} \left( 1/m_a \sum_{a_{k,i}=a} l(h(x_{k,i}), y_{k,i}) - \zeta/K \right) \leq 0.$$

**Benefits of BGL.** BGL ensures that the prediction quality on any group reaches a certain threshold. Compared to standard loss parity constraints that aim to equalize the losses across all protected groups (e.g. overall accuracy equity (Dieterich et al., 2016)), BGL has two main advantages. First, $G(w; \boldsymbol{\lambda})$ preserves convexity in $w$, as long as the loss function $l$ itself is convex. In contrast, loss parity constraints are generally non-convex even if $l$ is convex. Second, when the prediction difficulties are uneven across groups, loss parity may force an unnecessary drop of accuracy on some groups just to equalize all losses (Agarwal et al., 2019). In contrast, the criterion of BGL can avoid such undesirable effects.

**Instantiation 2 (Conditional Bounded Group Loss).** In some applications one needs a stronger fairness notion beyond ensuring that no group's loss is too large. For example, in the scenario of binary classification, a commonly used fairness requirement is equalized true positive rate or false positive rate (Hardt et al., 2016). In the context of optimization for arbitrary loss functions, a natural substitute is equalized true / false positive loss. In other words, any group's loss conditioned on positively / negatively labeled data should be equalized. Therefore, similar to BGL, we propose a novel fairness definition known as *Conditional Bounded Group Loss (CBGL)* defined below:

**Definition 2.** *A classifier $h$ satisfies Conditional Bounded Group Loss (CBGL) for $y \in Y$ at level $\zeta_y$ under distribution $\mathcal{D}$ if for all $a \in A$, we have $\mathbb{E}\left[l(h(x), y) | A = a, Y = y\right] \leq \zeta_y$.*

In practice, we could define empirical Conditional Bounded Group Loss constraint at level $[\zeta_y]_{y \in Y}$ under $\widehat{\mathcal{D}}$ to be

$$\mathbf{r}_{a,y}(h) := \sum_{k=1}^{K} \mathbf{r}_{(a,y),k}(h) = \sum_{k=1}^{K} \left( 1/m_{a,y} \sum_{a_{k,i}=a, y_{k,i}=y} l(h(x_{k,i}), y_{k,i}) - \zeta_y/K \right) \leq 0.$$

Note that satisfying CBGL for all $Y$ is a strictly harder problem than satisfying BGL alone. In fact, we can show that a classifier that satisfies CBGL at level $[\zeta_y]_{y \in Y}$ also satisfies BGL at level $\mathbb{E}_{y \sim \rho_a}[\zeta_y]$ where $\rho_a$ be the probability density of labels for all data from group $a$.

**Relationship between CBGL and Equalized Odds.**   For binary classification tasks in centralized settings, a common fairness notion is Equalized Odds (EO) (Hardt et al., 2016), which requires the True/False Positive Rate to be equal for all groups. Our CBGL definition can be viewed as a relaxation of EO. Consider a binary classification example where $Y = \{0, 1\}$. Let the loss function $l$ be the 0-1 loss. CBGL requires classifier $h$ to satisfy $\Pr[h(x) = y|Y = y_0, A = a] \leq \zeta_{y_0}$ for all $a \in A$ and $y_0 \in Y$. EO requires $\Pr[h(x) = y|Y = y_0, A = a]$ to be the same for all $a \in A$ given a fixed $y_0$, which may not be feasible if the hypothesis class $\mathcal{H}$ is not rich enough. Instead of requiring equity of each group's TPR/FPR, CBGL only imposes an upper bound for each group's TPR/FPR. Similar to the comparison between BGL and loss parity, CBGL offers more flexibility than EO since it does not force an artificial increase on FPR or FNR when a prediction task on one of the protected groups is much harder. In addition, for applications where logistic regression or DNNs are used (e.g., CV, NLP), it is uncommon to use the 0-1 loss in the objective. Thus, CBGL can provide a relaxed notion of fairness for more general loss functions whose level of fairness can be flexibly tuned.

**Instantiation 3 (MinMax Fairness).**   Recently Papadaki et al. (2022) proposed a framework called FedMinMax by solving an agnostic fair federated learning framework where the weight is applied to empirical risk conditioned on each group. Note that using BGL as the fairness constraint, our framework could reduce to FedMinMax as a special case by setting $\beta = 0, B = 1$ and $\zeta = 0$.

**Definition 3.** *Use the same definition of $\mathbf{r}_a(h)$ as we had in Instantiation 1. FedMinMax (Papadaki et al., 2022) aims to solve for the following objective:* $\min_h \max_{\boldsymbol{\lambda} \in \mathbb{R}_+^{|A|}, \|\boldsymbol{\lambda}\|_1 = 1} \sum_{a \in A} \boldsymbol{\lambda}_a \mathbf{r}_a(h)$.

Note that a key property of FedMinMax is the constant $\zeta$ used to upper bound the per group loss is set to 0. From a constrained optimization view, the only feasible solution that satisfies all fairness constraints for this problem is a model with perfect utility performance since requiring all losses to be smaller than 0 is equivalent to having all of them to be exactly 0. Such a property limits the ability to provide fairness guarantees for FedMinMax. Fixing $B$ and $\zeta$ also limits its empirical performance on the relation between fairness and utility, as we will show later in Appendix F.

## 4   PROVABLY FAIR FEDERATED LEARNING

In this section, we first propose *Provably Fair Federated Learning (PFFL)*, a scalable solver for Main Objective, presented in Algorithm 1. We provide formal convergence guarantees for the method in Section 4.2. Given the solution found by PFFL, in Section 4.3 we then demonstrate the fairness guarantee for different examples of fairness notions defined in Section 3 (BGL, CBGL).

### 4.1   ALGORITHM

To find a saddle point for Main Objective, we follow the scheme from Freund & Schapire (1997) and summarize our solver for fair FL in Algorithm 1 (full algorithm description in Appendix A). Our algorithm is based off of FedAvg (McMahan et al., 2017), a common scalable approach in federated learning. Intuitively, the method alternates between two steps:

(1) given a fixed $\boldsymbol{\lambda}$, optimize our regularized objective $F(w) + \boldsymbol{\lambda}^T \mathbf{r}(w)$ over $w$;
(2) given a fixed $w$, optimize the fairness violation term $\boldsymbol{\lambda}^T \mathbf{r}(w)$ over $\boldsymbol{\lambda}$.

While Agarwal et al. (2019) also follows a similar recipe to ensure BGL, our method needs to overcome additional challenges in the federated settings. In particular, the method in Agarwal et al. (2019) optimizes $w$ by performing exact best response, which is in general in feasible when data for distributed data sets. Our method overcomes this challenge by applying a gradient-descent-ascent style optimization process that utilizes the output of a FL learning algorithm as an approximation for the best response. In Algorithm 1, we provide an example in which the first step is achieved by using FedAvg to solve $\min_w F(w) + \boldsymbol{\lambda}^T \mathbf{r}(w)$ (L 4-12). Note that solving this objective does not require the FedAvg solver; any algorithm that learns a global model in FL could be used to find a certain $w$ given $\boldsymbol{\lambda}$. After we obtain a global model from a federated training round, we use exponentiated gradient descent to update $\boldsymbol{\lambda}$, following Alg 2 in Agarwal et al. (2019). This completes one training round. At the end of training, we calculate and return the average iterate as the fair global model.

Note that the ultimate goal to solve for Main Objective is to find a $w$ such that it minimizes the empirical risk subject to $\mathbf{r}(w) \leq 0$. Therefore, at the end of training, our algorithm checks whether

the resulting model $\bar{w}$ violates the fairness guarantee by at most some constant error $\frac{M+2\nu}{B}$ where $M$ is the upper bound for the empirical risk and $\nu$ is the upper bound provided in Equation 5 (L 16-20). We will show in the Lemma 1 that this is always true when there exists a solution $w^*$ for Problem 3. However, it is also worth noting that the Problem 3 is not always feasible. For example when we set $\zeta = 0$, requiring $\mathbf{r}(w) \leq 0$ is equivalent to requiring the empirical risk given any group $a \in A$ is non positive, which is only feasible when the loss is 0 for every data in the dataset. In this case, our algorithm will simply output *null* if the fairness guarantee is violated by an error larger than $\frac{M+2\nu}{B}$.

**Privacy aspect of PFFL** Compared to FedAvg, our solver communicates losses conditioned on each group in addition to model updates. This is common in prior works that solve a min-max optimization problem in federated learning (Hou et al., 2021; Zeng et al., 2021). We note that our method could be easily extended to satisfy example-level DP for FL by performing DP-SGD locally for each client. Our algorithm also gives natural client-level DP and LDP variants. In particular, we can compute via a trusted server the average loss at each client, which is sufficient statistics to update $\lambda$.

## 4.2 CONVERGENCE GUARANTEE

Different from Agarwal et al. (2019), while our algorithm handles arbitrary convex losses in federated setting by replacing the best response with the FedAvg output, we want to show that after running finitely many rounds, how close our solution is to the actual best response. In this section, we provide a no regret bound style analysis for our PFFL algorithm. To formally measure the the distance between the solution found by our algorithm and the optimal solution, we introduce $\nu$-approximate saddle point as a generalization of saddle point (See Remark in Section 3.1) defined below:

**Definition 4.** $(\widehat{w}, \widehat{\boldsymbol{\lambda}})$ is a $\nu$-approximate saddle point of $G$ if

$$
\begin{aligned}
G(\widehat{w}, \widehat{\boldsymbol{\lambda}}) &\leq G(w, \widehat{\boldsymbol{\lambda}}) + \nu \quad \text{for all } w \\
G(\widehat{w}, \widehat{\boldsymbol{\lambda}}) &\geq G(\widehat{w}, \boldsymbol{\lambda}) - \nu \quad \text{for all } \boldsymbol{\lambda}
\end{aligned}
\tag{4}
$$

As an example, the optimal solution $(w^*, \boldsymbol{\lambda}^*)$ is a 0-approximate saddle point of $G$. To show convergence, we first introduce some basic assumptions below:

**Assumption 1.** *Let $f_k$ be $\mu$-strongly convex and $L$-smooth for all $k = 1, \cdots, K$.*

**Assumption 2.** *Assume the stochastic gradient of $f_k$ has bounded variance: $\mathbb{E}[\|\nabla f_i(w_t^k; \xi_t^k) - \nabla f_k(w_t^k)\|^2] \leq \sigma_k^2$ for all $k = 1, \cdots, K$.*

**Assumption 3.** *Assume the stochastic gradient of $f_k$ is uniformly bounded: $\mathbb{E}[\|\nabla f_k(w_t^k; \xi_t^k)\|^2] \leq G^2$ for all $k = 1, \cdots, K$.*

These are common assumptions used when proving the convergence for FedAvg (e.g., Li et al., 2019b). Now we present our main theorem of convergence:

**Theorem 1** (Informal Convergence Guarantee). *Let Assumption 1-3 hold. Define $\kappa = \frac{L}{\mu}$, $\gamma = \max\{8\kappa, J\}$ and step size $\eta_Q = \frac{2}{(\beta+B)\mu(\gamma+t)}$, and assume $\|\mathbf{r}\|_\infty \leq \rho$. Letting $\bar{w} = \frac{1}{ET}\sum_{t=1}^{ET} w^t$, $\bar{\boldsymbol{\lambda}} = \frac{1}{ET}\sum_{t=1}^{ET} \boldsymbol{\lambda}^t$, we have:*

$$
\max_{\boldsymbol{\lambda}} G(\bar{w}; \boldsymbol{\lambda}) - \min_w G(w; \bar{\boldsymbol{\lambda}}) \leq \frac{1}{T}\sum_{t=1}^{T} \frac{\kappa}{\gamma+t-1}C + \frac{B\log(Z+1)}{\eta_\theta ET} + \eta_\theta \rho^2 B,
\tag{5}
$$

*where $C$ is a constant.*

The upper bound in Equation 5 consists of two parts: (1) the error for the FedAvg process to obtain $\bar{w}$ which is a term of order $\mathcal{O}(\log T/T)$; (2) the error for the Exponentiated Gradient Ascent process to obtain $\bar{\boldsymbol{\lambda}}$ which converges to a noise ball given a fixed $\eta_\theta$. Following Theorem 1, we could express the solution of Algorithm 1 as a $\nu$-approximate saddle point of $G$ by picking appropriate $\eta_\theta$ and $T$:

**Corollary 2.** *Let $\eta_\theta = \frac{\nu}{2\rho^2 B}$ and $T \geq \frac{1}{\nu(\gamma+1)-2\kappa\mathcal{C}}\left(\frac{4\rho^2 B^2 \log(Z+1)(\gamma+1)}{\nu E} + 2\kappa\mathcal{C}(\gamma-1)\right)$, then $(\bar{w}, \bar{\boldsymbol{\lambda}})$ is a $\nu$-approximate saddle point of $G$.*

We provide detailed proofs for both Theorem 1 and Corollary 2 in Appendix C. Different from the setting in prior FedAvg analyses (e.g., Li et al., 2019b), in our case the outer minimization problem changes as $\boldsymbol{\lambda}$ gets updated. Therefore, our analysis necessitates considering a more general scenario where the objective function could change over time.

## 4.3 FAIRNESS GUARANTEE

In the previous section, we demonstrated that our Algorithm 1 could converge and find a $\nu$-approximate saddle point of the objective $G$. In this section, we further motivate why we care

about finding a $\nu$-approximate saddle point. The ultimate goal for our algorithm is to: (1) learn a model that produces fair predictions on training data, and (2) more importantly, produces fair predictions on test data, i.e., data from federated clients not seen during training.

Before presenting the formal fairness and generalization guarantees, we state the following additional assumption, which is a common assumption for showing the generalization guarantee using the Rademacher complexity generalizations bound (Mohri et al., 2019).

**Assumption 4.** *Let $\mathcal{F}$ and $F$ be upper bounded by constant $M$.*

We first show the fairness guarantee on the training data.

**Lemma 1** (Empirical Fairness Guarantee). *Assume there exists $w^*$ satisfies $\mathbf{r}(w^*) \leq \mathbf{0}_Z$, we have*

$$\max_j \mathbf{r}_j(\bar{w})_+ \leq \frac{M + 2\nu}{B}. \tag{6}$$

Lemma 1 characterizes the upper bound for the worst fairness constraint evaluated on the training data. Given a fixed $\nu$, one could increase $B$ to obtain a stronger fairness guarantee, i.e. a smaller upper bound. Combining this with Corollary 2, it can be seen that when $B$ is large, additional exponentiated gradient ascent rounds are required to achieve stronger fairness.

Next we formalize the fairness guarantee for the entire true data distribution. Define the true data distribution to be $\mathcal{D} = \frac{1}{K} \sum_{k=1}^{K} \mathcal{D}_k$. We would like to formalize how well our model is evaluated on the true distribution $\mathcal{D}$ as well as how well the fairness constraint is satisfied under $\mathcal{D}$. This result is presented below in Theorem 3.

**Theorem 3** (Full Fairness and Generalization Guarantee). *Let Assumption 1-4 holds and $(\bar{w}, \bar{\lambda})$ a $\nu$-approximate saddle point of $G$. Then with probability $1 - \delta$, either there doesn't exist solution for Problem 3 and Algorithm 1 returns null or Algorithm 1 returns $\bar{w}$ satisfies*

$$\begin{aligned}
\mathcal{F}(\bar{w}) &\leq \mathcal{F}(w^*) + 2\nu + 4\mathfrak{R}_m(\mathcal{H}) + \frac{2M}{K} \sqrt{\sum_{k=1}^{K} \frac{1}{2m_k} \log(2/\delta)}, \\
\mathfrak{r}_j(\bar{w}) &\leq \frac{M+2\nu}{B} + Gen_{\mathbf{r},j}
\end{aligned} \tag{7}$$

*where $w^*$ is a solution for Problem 3 and $Gen_{\mathbf{r}}$ is the generalization error.*

The first part for Equation 7 characterizes how well our model performs over the true data distribution compared to the optimal solution. As number of clients $K$ increases, we achieve smaller generalization error. The second part for Equation 7 characterizes how well the fairness constraints are satisfied over the true data distribution. Note that the upper bound could be viewed as the sum of empirical fairness violation and a generalization error. Based on our fairness notions defined in Section 3.2, we demonstrate what generalization error is under different fairness constraints $\mathbf{r}$.

**Proposition 1** (**r** encodes BGL at level $\varsigma$). There are in total $|A|$ fairness constraints, one for each group. Define the weighted rademacher complexity for group $a$ as $\mathfrak{R}_a(\mathcal{H}) = \mathbb{E}_{S_k \sim \mathcal{D}_k^{m_k}, \sigma} \left[ \sup_{h \in \mathcal{H}} \sum_{k=1}^{K} \frac{1}{m_a} \sum_{a_{k,i}=a} \sigma_{k,i} l\left(h(x_{k,i}), y_{k,i}\right) \right]$. In this scenario, we have:

$$Gen_{\mathbf{r},a} = 2\mathfrak{R}_a(\mathcal{H}) + \frac{M}{m_a} \sqrt{\frac{K}{2} \log(2|A|/\delta)}.$$

Note that the fairness constraint for group $a$ under true distribution in Equation 7 is upper bounded by $\mathcal{O}\left(\frac{\sqrt{K}}{m_a}\right)$. For any group $a_0$ with sufficient data, i.e., $m_{a_0}$ is large, the BGL constraint with respect to group $a_0$ under $\mathcal{D}$ has a stronger formal fairness guarantee compared to any group with less data. It is also worth noting that this generalization error grows as the number of clients $K$ grows. Recall that the generalization error becomes smaller when $K$ grows; combing the two results together provides us a tradeoff between fairness notion of BGL and utility over the true data distribution in terms of $K$.

**Proposition 2** (**r** encodes CBGL at level $[\varsigma_y]_{y \in Y}$). There are in total $|A||Y|$ fairness constraints, one for each group and label. Define the weighted rademacher complexity for group $a$ conditioned on $y$ as $\mathfrak{R}_{a,y}(\mathcal{H}) = \mathbb{E}_{S_k \sim \mathcal{D}_k^{m_k}, \sigma} \left[ \sup_{h \in \mathcal{H}} \sum_{k=1}^{K} \frac{1}{m_{a,y}} \sum_{a_{k,i}=a, y_{k,i}=y} \sigma_{k,i} l\left(h(x_{k,i}), y\right) \right]$ where $m_{a,y}$ is the number of all examples from group $a$ with label $y$. In this scenario, we have:

$$Gen_{\mathbf{r},(a,y)} = 2\mathfrak{R}_{a,y}(\mathcal{H}) + \frac{M}{m_{a,y}} \sqrt{\frac{K}{2} \log(2|A||Y|/\delta)}.$$

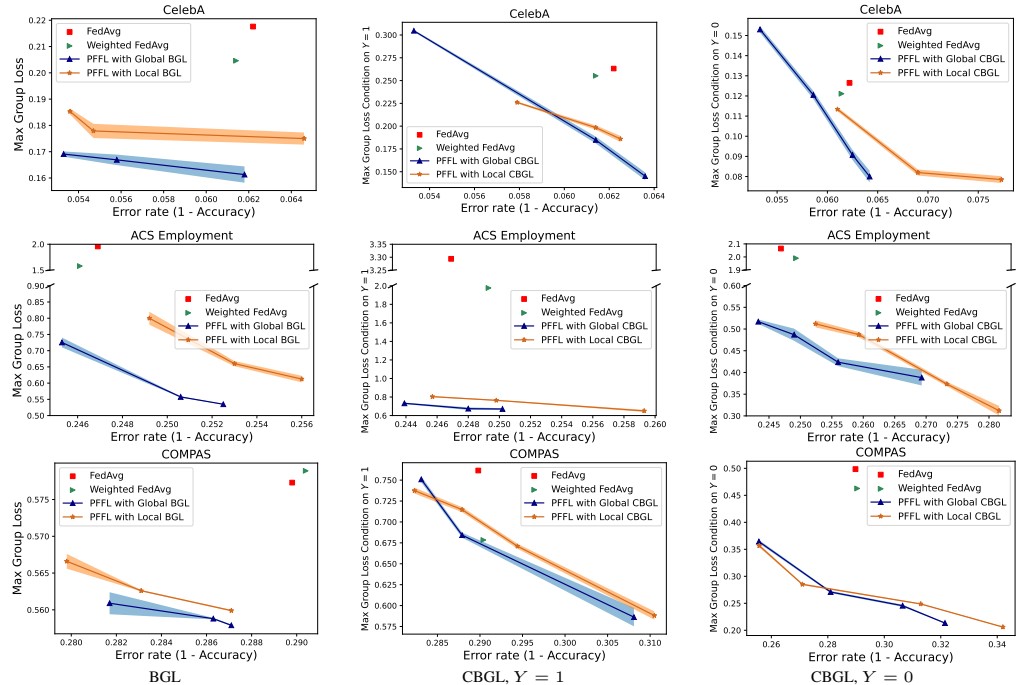

**Figure 2:** Experimental results for using BGL (column 1), CBGL for $Y = 1$ (column 2), and CBGL for $Y = 0$ (column 3) on CelebA (row 1), ACS Employment (row 2), and COMPAS (row 3). Interestingly, we find in all settings that our proposed method (PPFL with Global BGL) not only enables a flexible fairness/utility trade-off, but can in fact achieve both stronger fairness and better utility (lower error) than baselines.

Similar to Proposition 1, in order to achieve strong fairness guarantees for any specific constraint on the true data distribution, we need a sufficient number of samples associated with that constraint.

We provide details and proof for Theorem 3 in Appendix D. Different from the analysis performed in Agarwal et al. (2019), we analyze the generalization behaviour in federated setting where we introduce the generalization bound as a function of number of clients $K$. We then further formally demonstrate the tension between utility and fairness performance evaluated on the true data distribution induced by $K$, which has not been studied previously to the best of our knowledge.

## 5 EXPERIMENTS

We evaluate PFFL (Algorithm 1) empirically on ProPublica COMPAS, a dataset commonly studied in fair ML (Angwin et al., 2016; Zeng et al., 2021); the US-wide ACS PUMS data, a recent group fairness benchmark dataset (Ding et al., 2021); and CelebA (Caldas et al., 2018), a common federated learning dataset. We compare our method with training a vanilla FedAvg model in terms of both fairness and utility in Section 5.1, and explore performance relative to baselines that aim to enforce other fairness notions (Demographic Parity and Equal Opportunity) in Section 5.2.

**Setup.** For all experiments, we evaluate the accuracy and the empirical loss for each group on test data that belongs to all the silos of our fair federated learning solver. We consider COMPAS Recidivism prediction with gender as protected attribute, the ACS Employment task (Ding et al., 2021) with race as protected attribute, and CelebA (Caldas et al., 2018) with gender as a protected attribute. To reflect the federated setting, we use heterogeneous data partitions to create data silos. ACS Employent is naturally partitioned into 50 states; COMPAS and CelebA are manually partitioned in a non-IID manner into a collection of data silos. A detailed description of datasets, models, and partition schemes can be found in Appendix B.

### 5.1 FAIRNESS-UTILITY TRADE-OFFS FOR ALGORITHM 1

We first explore how test error rate differs as a function maximum group loss using our Algorithm 1. To be consistent with our method and theoretical analysis, we exclude the protected attribute $a_i$ for each data as a feature for learning the predictor. For each dataset, we evaluated PFFL with BGL; CBGL for $Y = 1$; and CBGL for $Y = 0$. For each method we evaluate, given fixed number of training iterations $E$ and $T$, we finetune $B$ and $\zeta$ and evaluate both test error rate and test loss on each

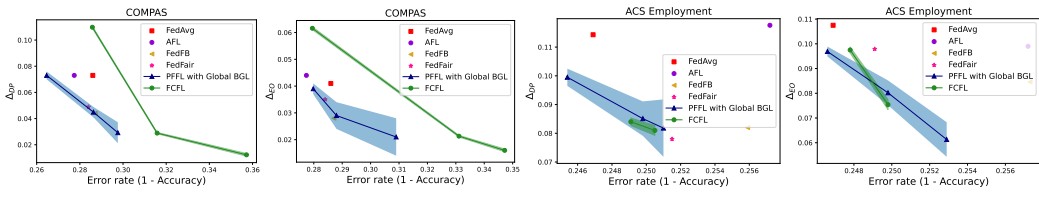

**Figure 3:** Comparison between PFFL and three different prior works on COMPAS and ACS for $\Delta DP$ and $\Delta EO$. Although PPFL was not directly designed to optimize Demographic Parity/Equal Opportunity, we see that it outperforms the baseline of FedAvg, and performs comparably to / better than prior works that were designed for these objectives.

group. Given a certain test error rate, we select the hyperparameter pair $(B, \zeta)$ that yields the lowest maximum group loss. We show the relation between test accuracy vs. max group loss in Figure 2. In particular, we compare our fairness-aware FL methods with two baseline methods: vanilla FedAvg and FedAvg trained on loss weighted by groups. In FL, applying fair training locally at each data silo and aggregating the resulting model may not provide strong population-wide fairness guarantees with the same fairness definition (Zeng et al., 2021). Hence, we also explore the relationship between test accuracy and max group loss under local BGL and global BGL constraints.

On all datasets, there exists a natural tradeoff between error rate and the fairness constraint: when a model achieves stronger fairness (smaller max group loss), the model tends to have worse utility (higher error rate). However, in all scenarios, our method not only yields a model with significantly smaller maximum group loss than vanilla FedAvg, but also achieves higher test accuracy than the baseline FedAvg which is unaware of group fairness. Meanwhile, for all datasets and fairness metrics, as expected, PFFL with Global BGL achieves improved fairness-utility tradeoffs relative to PFFL with Local BGL. Therefore, our PFFL with Global fairness constraint framework yields a model where utility can coexist with fairness constraints relying on Bounded Group Loss.

### 5.2 BGL/CBGL EVALUATED ON OTHER FAIRNESS NOTIONS

Beyond BGL and CBGL, there are other fairness notions commonly used in the fair machine learning literature. For example, several works in group fair FL have proposed optimizing the difference between every two groups' losses (possibly conditioned on the true label) with the aim of achieving Demographic Parity (or Equal Opportunity) (Chu et al., 2021; Cui et al., 2021; Hardt et al., 2016; Zeng et al., 2021). Formally, consider the case where the protected attribute set $A = \{0, 1\}$. Define $\Delta_{DP} = |\Pr(h(X) = 1|A = 0) - \Pr(h(X) = 1|A = 1)|$ and $\Delta_{EO} = |\Pr(h(X) = 1|A = 0, Y = 1) - \Pr(h(X) = 1|A = 1, Y = 1)|$. These works aim to train a model such that we could achieve small $\Delta_{DP}$ or small $\Delta_{EO}$, depending on the fairness constraint selected during optimization. As discussed in Section 3.2, CBGL could be viewed as a more general definition of Equal Opportunity and Equalized Odds. In this section, we compare our method with FedFB (Zeng et al., 2021), FedFair (Chu et al., 2021), and FCFL (Cui et al., 2021), all of which aim to optimize $\Delta_{DP}$ and $\Delta_{EO}$. We evaluate $\Delta_{DP}$ and $\Delta_{EO}$ for all approaches on COMPAS and ACS Employment, with results shown in Figure 3. Similar to Figure 2, we only show the points lying on the pareto frontier for our method. Although PFFL with BGL and CBGL was not directly designed for this fairness criteria (i.e., it does not directly enforce the loss or prediction parity of two groups' losses to be close), we see that our method is still able to outperform training a FedAvg baseline, and in fact performs comparably or better than prior methods (which were designed for this setting).

### 6 CONCLUSIONS, LIMITATIONS, AND FUTURE WORK

In this work, we propose a fair learning objective for federated settings via Bounded Group Loss. We then propose a scalable federated solver to find an approximate saddle point for the objective. Theoretically, we provide convergence and fairness guarantees for our method. Empirically, we show that our method can provide high accuracy and fairness simultaneously across tasks from fair ML and federated learning. In addition to strong empirical performance, ours is the first work we are aware of to provide formal convergence and fairness/generalization guarantees for group fair FL with general convex loss functions. In future work we are interested in investigating additional benefits that could be provided by using our framework, including applications in non-federated settings. Finally, similar to prior works in group fair FL, our method communicates additional parameters beyond standard non-fair FL (e.g., via FedAvg); studying and mitigating the privacy risks of such communications in the context of fair federated learning would be an interesting direction of future work.

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
