# OpenReview forum: "Fair Federated Learning via Bounded Group Loss"
_ICLR.cc/2023/Conference — Submitted to ICLR 2023_

### Official Review · Reviewer_KBz5 · 2022-10-24

**Confidence:** 3
**Correctness:** 2
**Technical Novelty And Significance:** 2
**Empirical Novelty And Significance:** 2
**Recommendation:** 3

**Clarity, Quality, Novelty And Reproducibility:**

- I'm not sure how Fig. 1 is supposed to be interpreted. I do not think it gives a clear motivation of the problem. I suggest the authors to either modify this figure (and caption) to make it more illustrative, or remove it all together. Instead, I suggest the authors to add Alg. 1 to the main paper, as opposed to leaving it entirely in the appendix.

**Details Of Ethics Concerns:**

As stated in the main review, I worry about the formulation of fairness and how it affects the direction of fairness ML. This formulation is exactly the type of (un)fairness that we as a society are moving away, i.e., setting the **SAME** threshold for **EVERY** group to satisfy, i.e., the same credit score for all groups. We should avoid this!

**Strength And Weaknesses:**

Strength:
+ Given bounded group loss (BGL) as the fairness criterion, this paper develops a FL solution that optimizes empirical risks for different groups under absolute loss constraints.
+ Theoretical analysis of convergence and fairness guarantee is provided.

Weakness:
- The biggest issue of this work is the fairness criterion that, in my opinion, violates the core principle of fairness. The BGL definition in Def. 1 uses an absolute average loss threshold for every (restricted) group $a \in A$. The authors' argument to move away from the equity between different groups and move towards such absolute, one-size-for-all quality threshold is highly flawed. First, the authors argue that this makes the problem convex, but such consideration is based on whether the authors can technically solve the problem as opposed to what is the right metric for fairness. Second, the argument that this metric "boosts the worst group's prediction quality" is solely focused on one group's performance, as opposed to the fairness **among all groups**. Fundamentally, fairness must consider all stakeholders. Last but probably the most importantly, using a single threshold could essentially lead to one group (the so-called worst group) barely meeting this threshold, while other groups achieve significantly smaller loss values. The authors consider this case to be fair, but in reality, such prediction outcome discrepancy can still lead to unfair decision making. Additionally, choosing such threshold $\zeta$ becomes non-trivial, as it significantly affects BGL. Having the algorithm fairness depend critically on hyperparameter tuning is dangerous and should be avoided when possible. As the whole work is built on such absolute threshold, I believe this is a critical issue that should be addressed.
- As the authors stated in the Remark, $G$ is linear in $\lambda$ and convex in $w_0$. I'm not sure why the optimal solution has to be defined w.r.t. a saddle point instead of the true optimum.
- The novelty of this work is limited given the state of the art. The algorithmic approach described in Sec. 4.1 is a standard alternative optimization procedure. The technique used in the convergence analysis in Sec. 4.2 is very similar to the prior approach, e.g., in Li et al. 2019b.

**Summary Of The Paper:**

This paper studies fair federated learning (FL) where each group of clients are guaranteed to have the same loss function value upper bound. Theoretical guarantees for both model convergence (for convex loss functions) and fairness are provided, together with experimental results that support the claims.

**Summary Of The Review:**

This work is technically sound given the premise of the fairness formulation. However, this premise does not seem to satisfy the core principle of fairness -- treating all groups fairly.

---

> ### Author Response · Authors · 2022-11-16
> **Response to Reviewer KBz5**
>
> We thank the reviewer for taking time to review our work and for your detailed comments. We provide answers to the points that were raised below:
>
> **BGL**: Thanks for the comment. First of all, we wish to reemphasize that BGL is an existing algorithmic fairness notion that has been studied and used in prior work in fair ML (please see response to all reviewers). In this work, we focus on leveraging BGL as a tool to unlock general group fair federated learning that comes with provable fairness guarantee (Theorem 3) in the context that groups are distributed across silos with heterogeneous data distribution instead of proposing BGL as a new fairness notion. The hope is that by using BGL and its variants, we may be able to sacrifice less utility while achieving fairness. Second, the fact that the metric ‘‘boosts the worst group's prediction quality’’ comes from the definition of BGL that requires *ALL* groups’ loss to be smaller than some certain threshold. Therefore, contrary to what the reviewer claims, BGL focuses on fairness of all groups (no group should perform worse than the other group by a certain constant), and as a consequence of that, the algorithm improves the worst group’s prediction quality rather than only focusing on optimizing for groups with less prediction difficulty.
>
> We also disagree with the reviewer on the concern that the proposed method would lead to the worst group “barely meeting this threshold, while other groups achieve significantly smaller loss values”. Our experimental results (Figure 3) suggests the proposed method effectively reduces the DP / EO gap (equity of prediction quality among groups) compared to fairness-unaware methods like FedAvg. More importantly, compared to baselines that are explicitly designed to optimize the DP / EO gap, our method achieves comparable / even stronger fairness-utility trade offs, even though we did not directly optimize for these criteria. For more details on the motivations of BGL, please see our response to all reviewers.
>
> **Figure 1 and Algorithm 1**: Thanks for your suggestions. Figure 1 explains why data heterogeneity would be a problem when solving the group fair FL problem: naively applying fair training locally at each silo might fail to produce a globally fair model since the local fairness statistics (e.g. acceptance rate, TPR/FPR, max group loss, etc.) could not represent global fairness statistics given heterogeneous distribution across clients. We will add more clarifications in our revision. We will also move the algorithm description to the main body in our revision.

---

> > ### Author Response · Authors · 2022-11-17
> > **Follow up with the discussion**
> >
> > Dear Reviewer,
> >
> > Thanks again for your detailed feedback. As we are approaching the end of the discussion period, we would like to follow up with you to check whether our responses and new results have adequately addressed your questions/concerns.
> >
> > **Can you let us know if we've addressed your concerns, or if you have further questions?** If you have additional questions, we would love to have the opportunity to discuss in more detail.
> >
> > Thanks, Authors of 2066

---

### Official Review · Reviewer_ScJT · 2022-10-25

**Confidence:** 4
**Correctness:** 4
**Technical Novelty And Significance:** 3
**Empirical Novelty And Significance:** 3
**Recommendation:** 6

**Clarity, Quality, Novelty And Reproducibility:**

Clarity: The writing is ok, but lacks some literature search for federated saddle point optimization.
Quality: The optimization approach is standard.
Novelty: The provable guarantee for fair federated learning is novel.

**Strength And Weaknesses:**

Strength:
- The topic of fair federated learning is relevant.
- The technical idea of reducing to federated saddle point optimization is natural.

Weaknesses:
- The selection of Bounded Group Loss is not very well motivated. To my understanding, this selection is due to its convexity instead of its practical interest. The advantage of the proposed algorithm in Bounded Group Loss is not surprising since the baselines are not defined for this fairness measure.
- Lack of discussion on techniques. The reduction to federated saddle point optimization is natural, and hence I expect to see a comparison with federated saddle point optimization algorithms in the literature. For example, [Hou et al., Efficient Algorithms for Federated Saddle Point Optimization, 2021] and [Shen et al., FedMM: Saddle Point Optimization for Federated Adversarial Domain Adaptation, 2021].

**Summary Of The Paper:**

The paper proposes a fair learning objective for federated settings via Bounded Group Loss. The authors propose a scalable federated solver to find an approximate saddle point for the objective. Theoretically, they provide convergence and fairness guarantees for the method. Empirically, they show that their method can provide high accuracy and fairness simultaneously across tasks from fair ML and
federated learning.



**Summary Of The Review:**

The paper proposes a fair learning objective for federated settings via Bounded Group Loss. The selection of Bounded Group Loss is not very well motivated. The writing is ok but lacks some literature search for federated saddle point optimization. Overall, I do not recommend acceptance.

---

> ### Author Response · Authors · 2022-11-16
> **Response to Reviewer ScJT:**
>
> We thank the reviewer for taking time to review our work and for your detailed comments. We provide answers to the points that were raised below:
>
> **Motivation of BGL**: We gently disagree with the reviewer regarding the discussion around BGL. As we stated in our general response to all authors, parity difference based fairness notions like DP/EO enforces equal prediction across all groups, which might drastically sacrifice the utility performance in order to achieve fairness. To avoid such undesirable behavior, BGL tries to enforce all groups’ performance to be at least as good as a certain threshold. It provides a more flexible fairness notion compared to DP/EO. In the context of FL, we demonstrate that theoretically our method provides formal fairness guarantees (Theorem 3), which is missing in prior work. Contrary what the reviewer mentions, since BGL does *NOT* directly optimize other fairness notions like DP/EO, it is surprising that our method could provide comparable / better fairness-utility tradeoff under these metrics (Figure 3).
>
> **Federated Saddle Point Optimization**: A key difference between our solver and solvers in prior works on federated saddle point optimization (e.g. Hou et al., Shen et al.) lies in that our solver outputs the average iterate, which has a formal fairness guarantee, whereas solvers in prior works output the last iterate. Therefore, prior solvers could not directly apply to our objective which aims to provide provable guarantees.

---

> > ### Comment · Reviewer_ScJT · 2022-11-17
> > **Reply to the response**
> >
> > Thanks for the response. I think the empirical results that the proposed algorithms achieve better tradeoffs for DP & EO are fascinating, raising my score. By the way, I suggest the authors think about why this phenomenon happens, maybe intuition can motivate the selection of BGL better. Currently, I still think the motivation is not enough -- the fact that BGL appears in prior work does not ensure that it is worth studying in the federated setting.
> >
> > Also, I think the discussion of the technical novelty is still not convincing. Only mentioning that you output the average iterate still confuses me: What is the technical challenge of directly applying the prior solvers? Why does outputting the average iterate ensure fairness?

---

> > > ### Author Response · Authors · 2022-11-17
> > > **Response**
> > >
> > > Thank you again for raising the score and your comments:
> > >
> > > - As we discuss in our submission, the motivation for exploring BGL in federated settings stems from the fact that no existing solutions for fair FL can provide provable fairness guarantees. Such guarantees are not just something that are nice to have for a research paper---they have critical practical ramifications. In particular, as we see in our experiments, existing, heuristic approaches to fair FL may result in solutions that are not only subpar in terms of fairness/utility trade-offs, but may actually result in solutions that are less fair than enforcing no fairness at all (e.g., just running FedAvg on the standard ERM objective). In short, developing and understanding principled solutions to fair FL is crucial if we actually care about enforcing fairness in practice. That being said, we agree with the reviewer that further exploring BGL in the context of DP/EO (even outside of federated settings) would be an interesting direction of future work.
> > >
> > > - We consider the average iterate as it is standard when proving convergence of the dual variable $\lambda$ (see Shalev-Shwartz et al 2012 [1]) and thus allows us to use the $\nu$-approximate saddle point to bound the maximum empirical fairness violation (see Lemma 1 in Agarwal et al 2018 [2]), which is critical for providing our fairness guarantee in Theorem 3. It is not clear to us how to obtain such guarantees with the last iterate. Additionally, we note that a benefit of our proposed solver relative to prior work is that it considers an approximate best response for the model $w$ at each iteration, which is useful in practical federated settings. In particular, since we may be unable to find the best $w$ for a fixed dual variable $\lambda$ as we did in the centralized setting, we run enough numbers of FL training rounds (e.g. FedAvg in this work) to approximate the best response before updating the dual variable. Empirically, we find that such an update rule is beneficial for achieving better fairness-utility trade-offs in contrast to standard alternating update rules (e.g., as in FedMinMax; see Figure 6).
> > >
> > > [1]. Shai Shalev-Shwartz et al. Online learning and online convex optimization.Foundations and trends in Machine Learning, 4(2):107–194, 2011.
> > >
> > > [2]. Alekh Agarwal, Alina Beygelzimer, Miroslav Dudík, John Langford, and Hanna Wallach.A reductions approach to fair classification. In International Conference on Machine Learning. PMLR, 2018.

---

### Official Review · Reviewer_SyJQ · 2022-10-27

**Confidence:** 4
**Clarity, Quality, Novelty And Reproducibility:** The paper is clearly written.
**Correctness:** 2
**Technical Novelty And Significance:** 2
**Empirical Novelty And Significance:** 2
**Recommendation:** 3

**Strength And Weaknesses:**

-Privacy leakage/secrecy of the proposed algorithm is not discussed. It is unclear how much privacy leakage occurs. The proposed algorithm requires the exchange of additional information "r" per round, so one *must* discuss the implication of this. Federated learning algorithms with frequent synchronization/additional data exchange may be subject to an unacceptable level of information leakage, making it no different from sharing all the local datasets with every client. To avoid this kind of pitfall, one must clearly discuss the privacy aspect of new algorithms.

-BGL seems to be ok if it's used for regression or used for classification without any preferred labels. ACS Emp./COMPAS are the tasks with preferred outputs, so I am not sure if using BGL makes sense in these settings.

-Some of the experimental results look weird. In most of the experiments, FedAvg, which should minimize the error rate better than any other constrained counterparts, does not seem to achieve the lowest error rate. Sometimes it performs the worst among all tested methods. Why?

-How are the hyperparameters chosen for the baseline methods?

-Some missing baselines. Also, please add the performance of the standard reduction method with central data (upper bound).

[1] FairFed: Enabling Group Fairness in Federated Learning https://arxiv.org/abs/2110.00857
[2] GIFAIR-FL: A Framework for Group and Individual Fairness in Federated Learning https://arxiv.org/abs/2108.02741

**Summary Of The Paper:**

The paper proposes a group federated learning algorithm with formal convergence/BGL guarantees.

**Summary Of The Review:**

See my comments above.

---

> ### Author Response · Authors · 2022-11-16
> **Response to Reviewer SyJQ**
>
> We thank the reviewer for taking time to review our work and for your detailed comments. We provide answers to the points that were raised below:
>
> **Privacy**: As we mentioned in Section 6, the intersection between privacy and group fairness is an interesting direction that has not been formally explored in FL to the best of our knowledge. The main focus of this work is not to consider privacy or privacy-fairness trade-offs. First of all, in the context of differentially private federated learning (either client-level or example-level DP / LDP), compared to the model update communicated during training, which is a vector in $\mathbb{R}^d$, the privacy leakage of the loss (a scalar in $\mathbb{R}$) is very small. However, we note that our method could be easily extended to satisfy example-level DP for FL by performing DP-SGD locally for each client. Our algorithm also gives natural client-level DP and LDP variants. In particular, we can compute via a trusted server the average loss at each client, which is sufficient statistics to update $\lambda$. We will expand on this discussion in our revision (in particular, right after the solver is introduced) as per your suggestion.
>
> **Notion of BGL**: Thanks for bringing up this comment. Actually, the extension to CBGL exactly solves the reviewer’s question. By conditioning on the true label, CBGL enables each label to have different tolerance ($\zeta_y$) for maximum fairness violation. Therefore, based on the preferred labels, one could set smaller $\zeta_y$ to enforce stronger fairness.
>
> **Hyperparameters for baseline methods**: For baseline methods, we set the same learning rate and batch size as we did in the experiment of PFFL. For methods that come with additional hyperparameters (e.g. FedFB, FCFL), we choose the hyperparameter from the set of hyperparameters suggested by the original paper ($\epsilon$ in FedFB, fairness budget eps_g in FCFL) and plot the one that yields the best fairness given fixed utility.
>
> **Other baselines**: Thanks for suggesting these baselines. We didn’t compare to GIFAIR-FL as it is focusing on a different setting where a group is defined as a group of clients, whereas in our case we look at a more general scenario where a group consists of samples that could be distributed across clients. We also plot the comparison with FairFed in Figure 6 of Appendix H (Please see the full paper with appendix in the supplementary material). Our method achieves comparable / better results on both CelebA and COMPAS datasets. It is also worth noting that the nature of FairFed only supports two different groups (to calculate the weights) while our framework could be applied to scenarios where there are more than 2 groups.

---

> > ### Author Response · Authors · 2022-11-17
> > **Follow up with the discussion**
> >
> > Dear Reviewer,
> >
> > Thanks again for your detailed feedback. As we are approaching the end of the discussion period, we would like to follow up with you to check whether our responses and new results have adequately addressed your questions/concerns.
> >
> > **Can you let us know if we've addressed your concerns, or if you have further questions?** If you have additional questions, we would love to have the opportunity to discuss in more detail.
> >
> > Thanks, Authors of 2066

---

### Official Review · Reviewer_juWF · 2022-10-29

**Confidence:** 4
**Clarity, Quality, Novelty And Reproducibility:** The paper is clearly written and easi…
**Correctness:** 3
**Technical Novelty And Significance:** 2
**Empirical Novelty And Significance:** 2
**Recommendation:** 3

**Strength And Weaknesses:**

Strengths:

The paper allows the direct specification of target group-specific loss minimums, which are a natural way of specifying group constraints. The resulting algorithm is simple and well grounded, and the resulting fairness-utility tradeoffs seem comparable to existing approaches.

Weaknesses:

I think the novelty of BGL is limited, the proposed objective is equivalent to FedMinMax, which is discussed on the paper,  with an additional static bias added to the group-conditional loss terms. The formulation as a saddle point presented in this paper is not fully compatible with the bounds on the Lagrangian weights presented in Eq 2, and Algorithm 1 makes it seem like the weights asymptotically reach the norm bound B, which begs the question on why is that a separate optimization parameter from \beta in Eq 2.

The theoretical comparisons with existing methods, in particular group DRO, which uses a similar exponential weight update rule for lambda and thresholds on the group losses (albeit with an additional relu), and FedMinMax, which in its paper formulation has a lower bound constraint on lambda  which makes both objectives theoretically equivalent, is sorely lacking.


**Summary Of The Paper:**

The paper discusses the use of bounded group loss (BGL) in the context of fair federated learning. The core objective is to learn a classifier that satisfies some predefined constraints on the expected loss function of each group on average across all clients. It is expected that the client-conditional data distribution may differ between clients.

The paper further assumes that the bounded loss function is convex w.r.t. the model parameters, and proposes a distributed saddle point optimization algorithm to recover a model that minimizes empirical risk subject to these group constraints. A convergence proof for the algorithm is also provided


**Summary Of The Review:**

The proposed method is sensible, but there is limited novelty, and uncertainty if the method is not equivalent to existing approaches as it stands.

---

> ### Author Response · Authors · 2022-11-16
> **Response to Reviewer juWF**
>
> We thank the reviewer for taking time to review our work and for your detailed comments. We provide answers to the points that were raised below:
>
> **Comparing theoretical results**: Thanks for your suggestion! Could the reviewer please provide references on theoretical results for group DRO that they would like us to compare to? (E.g., do you wish to see fairness guarantees, convergence results, generalization guarantees, something else?) A key difference between our formulation and the group DRO objective in [1] is that we formulate the problem as a constrained optimization problem where the goal is to find a solution that satisfies the fairness constraints, whereas the group DRO objective simply aims to minimize the worst group’s performance. If the reviewer could point to any specific theoretical results in prior group DRO literature we would be happy to provide further discussion.
>
> [1]. Sagawa, S., Koh, P. W., Hashimoto, T. B., & Liang, P., Distributionally robust neural networks for group shifts: On the importance of regularization for worst-case generalization, ICLR 2020.

---

> > ### Author Response · Authors · 2022-11-17
> > **Follow up with the discussion**
> >
> > Dear Reviewer,
> >
> > Thanks again for your detailed feedback. As we are approaching the end of the discussion period, we would like to follow up with you to check whether our responses and new results have adequately addressed your questions/concerns.
> >
> > **Can you let us know if we've addressed your concerns, or if you have further questions?** If you have additional questions, we would love to have the opportunity to discuss in more detail.
> >
> > Thanks, Authors of 2066

---

### Author Response · Authors · 2022-11-16
**Response to all reviewers**

We thank all reviewers for their feedback and suggestions. We have updated our paper and appendix in the revision to reflect this feedback, with new changes highlighted in red. We clarify a shared concern amongst reviewers below:


**Novelty of BGL**: In this work we leverage BGL for fair FL as an alternative to fairness notions like DP/EO as a principled way to achieve group fairness in federated learning. We do *NOT* claim to propose BGL as a new fairness notion (BGL was first used in the context of fair ML in Agarwal et al. 2018). Compared to standard loss parity-based constraints used in prior works in fair FL, BGL has two major benefits in terms of theoretical guarantees and empirical performance: (1) BGL preserves convexity and smoothness of the objective, (2) BGL avoids drastically sacrificing utility performance when enforcing equity of different groups’ losses. As we observe in our experiments, these theoretical benefits also lead to the BGL objective being more well-behaved in practice—yielding fairness/utility benefits across all benchmarks we explore.

Although BGL has been used previously in the fairness literature, it was unclear how to solve the objective at scale in federated networks, and how to extend the theoretical results to federated settings. Key contributions of this work include providing provable fairness guarantees (Theorem 3) and a scalable solver (along with convergence guarantees) for group fair federated learning via BGL. Finally, we demonstrate for the first time strong practical benefits of using BGL: even though BGL does not directly optimize for DP/EO, our method provides comparable / better empirical results compared to all prior baselines along these standard metrics (Figure 3). Our method thus achieves strong performance both in theory and in practice, and we hope it will be a useful tool for future work in fair federated learning.

---

### Decision · Program_Chairs · 2023-01-20

**Decision:**

Reject

**Justification For Why Not Higher Score:**

See the aforementioned general concern, and the unclear motivation of the work and its relation to the other fairness notions in the context of FL.

**Justification For Why Not Lower Score:**

N/A.

**Metareview: Summary, Strengths And Weaknesses:**

The main contribution of this work lies in using bounded group loss for achieving group fairness in federated learning with provable fairness guarantees.

Such a fairness notion BGL is proposed by Agarwal et al. (2019) and relatively well received, albeit in a context different from FL. To date, there is no one fairness notion that "rules" them all, hence resulting in several fairness notions being proposed. So, at this time, we should remain open to the consideration of various fairness notions, including BGL.

After reviewing the authors' rebuttal and an active discussion, a general concern shared by most reviewers is that it is not immediately clear (beyond the theoretical guarantees and empirical results) why real-world applications need to consider BGL (over other fairness notions) specifically in the context of FL. Furthermore, the authors have mentioned in their rebuttal: "More importantly, compared to baselines that are explicitly designed to optimize the DP / EO gap, our method achieves comparable / even stronger fairness-utility trade offs, even though we did not directly optimize for these criteria." No explanation was given for the observation of this result, begging the question regarding the relationship of BGL to the other evaluated fairness notions in the context of FL. The above seems to have inspired the reviewers to ask various questions regarding the motivation of this work and its relation to the other fairness notions in the context of FL.

The authors are strongly encouraged to revise their work based on the above feedback and the reviewers' suggestions.